# Discontinuation of Antidepressants and the Risk of Medication Resumption among Community-Dwelling Older Adults with Depression in the US

**DOI:** 10.3390/ijerph21091209

**Published:** 2024-09-13

**Authors:** Easter P. Gain, Xinhua Yu, Satish K. Kedia, Abu Mohd Naser, Morgan I. Bromley, Mark’Quest Ajoku, Xichen Mou

**Affiliations:** 1Division of Epidemiology Biostatistics and Environmental Health, School of Public Health, University of Memphis, Memphis, TN 38152, USA; egain@memphis.edu (E.P.G.); atitu@memphis.edu (A.M.N.); mbromley@memphis.edu (M.I.B.); majoku@memphis.edu (M.A.); xmou@memphis.edu (X.M.); 2Division of Social and Behavioral Sciences, School of Public Health, University of Memphis, Memphis, TN 38152, USA; skkedia@memphis.edu

**Keywords:** antidepressants discontinuation, over-prescription, short duration, older adults

## Abstract

Antidepressants are among the most prescribed drugs in the US, but the current treatment patterns and modalities among older adults are unclear. This study assessed the patterns of discontinuation of antidepressants and the risk of medication resumption among community-dwelling older adults with depression. Using Medicare Current Beneficiary Survey (MCBS) data from 2015–2019, we identified 1084 beneficiaries with depression who newly initiated serotonergic antidepressants. The risk of medication resumption was explored using survival analysis. The median duration of continuous medication was 90 days. However, about 30% of patients had a treatment duration of 30 days or shorter, 26% for 31–90 days, 15% for 91–180 days, and 30% for 181 days or more. After adjusting for all covariates, patients with less than 30 days of continuous medication were half as likely to resume the medications compared to those with 91–180 days (HR: 0.49 (95% CI: 0.37, 0.65)). Nearly one-third of older adults used an antidepressant medication for a short duration with a lower risk of medication resumption. A shorter treatment duration without resumption might suggest over-prescription of antidepressants among community-dwelling older adults.

## 1. Introduction

Depression is the third leading cause of disability worldwide [1] and affects 15–18% of people throughout their lifetimes [2]. For example, the self-reported prevalence of lifetime diagnosis of depression was 14.2% among US adults aged 65 or older in the year 2020 [3]. The high prevalence of depression, compounded by the chronic nature of depression and co-existence with other morbidities, resulted in significant personal, economic, and public health consequences [4]. Meanwhile, treatments for depression have been evolving rapidly. The serotonergic antidepressants, often as the first-line treatment, were effective for about two-thirds of patients with moderate to severe depression [5,6]; and the use of antidepressants increased substantially among older adults in the US, rising from 15.3% in 2006–2007 to 23.9% in 2014–2015 [7]. On the other hand, the use of behavioral interventions, counseling, and other psychotherapies was not common among older adults with depression [8], leading to some serious concerns about antidepressant over-prescription [9,10].

Accumulating evidence shows that treatment guidelines for antidepressant medication are frequently not followed. A structured three-phase strategy for managing antidepressant medication is advocated for by the American Psychiatric Association (APA) to sustain long-term remission and minimize the risk of recurrence [5]. Specifically, the acute phase of treatment should last for at least 6–8 weeks, followed by a continuous phase lasting for 4–9 months and an extended maintenance phase for six months or more if needed. In addition, treatment regimens vary considerably, and many antidepressants have similar effectiveness but with different side effects. Inappropriate treatments such as short-term use, early discontinuation, and intermittent medications can lead to a higher risk of recurrence of depression and resumption of medication [11,12]. For example, about 30–40% of patients discontinued antidepressant treatment before completing the acute phase of their treatment, sometimes within the initial weeks of treatment [13,14,15,16]. Additionally, patients often stopped the treatment only after one prescription [14]. Early discontinuation during the acute phase of treatment posed a greater risk of medication resumption in the future compared to continuous treatment lasting for six months to one year [12,17], and more than half of the users experienced relapse when their treatment duration was shorter than three months [17,18,19]. Thus, adhering to the treatment guidelines is crucial for achieving successful long-term outcomes in treating depression.

In addition, antidepressants are used, sometimes “off-label”, for managing many other health conditions [20]. For example, anxiety disorders are effectively managed by SSRIs, and routinely, duloxetine is used for relieving neuropathic pain and trazodone for alleviating insomnia [21,22]. Thus, an increasing use of antidepressants among older adults may also reflect the increased prescriptions for managing other conditions which may be used in the short term.

However, evidence is scarce for the duration of antidepressant medication, patterns of discontinuation, and risk of medication resumption among older US adults with depression. Previous studies primarily focused on comparing medication resumption or depression recurrence between individuals who continued versus discontinued medication during the acute phase among adults [14,19,23]. Furthermore, studies have often examined findings from individuals using antidepressants without specifically targeting those receiving medications for treating depression [18,19].

Therefore, the current study aims to (1) evaluate the patterns of the duration and discontinuation of antidepressants medication and the impact on the risk of medication resumption; and (2) examine the factors determining the discontinuation of antidepressant medication among community-dwelling older adults with depression. Our study will focus on serotonergic antidepressants, as previous research demonstrated that over 90% of the first-line pharmacologic treatment for depression is a monotherapy with serotonergic antidepressants [6]. Our findings will enhance our understanding of antidepressant utilization patterns in this population and identify factors contributing to the discontinuation and medication resumption, thereby informing strategies for more effective depression management among older adults.

## 2. Materials and Methods

### 2.1. Data Source

This study used Medicare Current Beneficiary Survey (MCBS) data from 2015 to 2019. MCBS is an ongoing survey that enrolls beneficiaries in the Fall each year, collects information four times annually, and follows participants for a maximum of four years, resulting in a rotating panel of three complete follow-up years with twelve interviews for each beneficiary over the follow-up period. MCBS collects thorough information on participants’ health status, healthcare utilization, health insurance coverage, medication usage, and socio-demographic characteristics [24].

### 2.2. Study Cohort

We created a cohort of Medicare beneficiaries aged ≥65 years at the time of enrollment. We included patients who had at least twelve months of follow-up, a depression diagnosis within six months of antidepressant medication use or a Patient Health Questionnaire (PHQ) score ≥ 10, and newly initiated medication (i.e., no antidepressant use within six months of enrollment, though they might have antidepressants medication six months before). We excluded participants who died within six months of Fall enrolment and who were taking antidepressant medications for other purposes without an indication of depression (e.g., only for anxiety, stress, pain, insomnia, or other mental health conditions). The depression diagnosis was based on International Classification of Disease-9 (ICD-9) and International Classification of Disease-10 (ICD-10) codes from physician, inpatient, and outpatient Medicare claims matched within 180 days of antidepressant prescriptions (Appendix A, Table A1 for a list of depression diagnosis codes). Participants who self-reported a PHQ-8 (excluding the suicide question in PHQ-9) score 10 or more at the baseline survey were also considered to have depression. Finally, we limited our cohort to participants who were taking Selective Serotonin Reuptake Inhibitor (SSRI), Serotonin and Norepinephrine Reuptake Inhibitor (SNRI), and/or Serotonin Antagonist and Reuptake Inhibitor (SARI) (Appendix B, Table A2 for a list of generic drug names used in this study). Serotonergic antidepressants are the first-line treatment for depression.

### 2.3. Measures of Continuous Duration and Resumption of Medications

We used prescription dates and days of medication supply on the prescription drug claims to calculate the duration of medication use and gaps between medications. For each prescription, the starting date of medication was the prescription date, and the end date of medication was the prescription starting date plus days of supply. We calculated the prescription gap as the interval between the current prescription date and the end date of the previous prescription. If the gap between two consecutive prescriptions was 14 days or less, we considered these two prescriptions as continuous prescriptions, that is, the prescription gap was not a true medication gap. The combined durations of these consecutive prescriptions with a gap of 14 days or less were one continuous duration of medications. We used the difference of 14 days, as most serotonergic antidepressants have a washout period of 1–2 weeks for switching to a different drug [25]. The continuous duration of medications was categorized into four groups: ≤30 days, 31–90 days, 91–180 days, and ≥181 days. In this study, we did not examine the switch of drugs, as we have already limited the drug class to SSRI, SNRI, and SARI.

Another outcome of our study was the resumption of medications. If the prescription gap between two consecutive prescriptions was more than 14 days, we considered it a resumption of antidepressant medication. The resumption of medications was classified according to the medication gap as within three months (15–90 days), between three to six months (91–180 days), and six months or longer (≥181 days).

### 2.4. Predictors

The predictors or independent variables included demographic, socio-economic, and health status and other health conditions. They were gender (male/female), age (≥65–74, 75–84, and ≥85 years), race (white, black, and others), education (less than high school diploma, high school diploma, and college degree or higher), income (<$15,000, $15,000–<$30,000, $30,000–<$50,000, ≥$50,000 per year), marital status (married or living with partner, and divorced/widowed/never married/other), comorbidity (1, 2, 3, 4 or more comorbidities). Comorbidities included neuropsychic diseases like dementia, Parkinson’s disease, Alzheimer’s diseases; psychotic disorder excluding depression; osteoporosis and bone fracture; paralysis; heart diseases; stroke; diabetes; asthma; rheumatoid arthritis or osteoarthritis; hypertension; and cancer. For analytic purposes, specific comorbidities of heart diseases, stroke, diabetes, and cancer were included in the models as individual covariables, as these health conditions are highly comorbid with depression.

### 2.5. Statistical Analysis

Descriptive statistics such as the means/standard deviations of continuous variables and frequency of the categorical variables were reported with a *t*-test and Chi-squared test for comparisons between different treatment duration groups, respectively. No multiple comparisons were adjusted. To adjust for covariables, multivariable logistic regressions were used to estimate the odds ratios of resumption of medications after having shorter medication durations, and Cox proportional hazard regressions were used to estimate the hazard ratios for the time to medication resumption. Model assumptions were examined by residual analysis (e.g., Schoenfield residuals for proportionality of hazards in Cox regressions). All statistical analyses were performed using SAS, 9.4 (SAS Institute, Inc., Cary, NC, USA).

## 3. Results

### 3.1. Basic Characteristics by Antidepressant Medication Duration

Among 14,258 survey participants from 2015 to 2019, we identified 3135 community-dwelling older adults who took any antidepressant medications during the period. After limiting to those who initiated the antidepressants at least six months after enrollment and had at least one year of follow-up, there were 1084 participants with depression who were on SSRI (691), SNRI (190), and SARI (203) medications.

Table 1 shows their socio-demographic and clinical characteristics, stratified by the duration of continuous prescription medications. Overall, about 30% of the patients had a continuous medication duration of more than six months (180+ days). However, about 29% of patients had a prescription duration of 30 days or less, and 26% of patients had a prescription duration of 31–90 days. The median days of continuous medication duration were 90 days.

Furthermore, antidepressant users were more likely to be female, younger (65 to 74), white, with income between 15K to 30K, have a college degree or above, be married or with partner, or have ≥4 comorbidities. More importantly, those aged 85+ were more likely to take medications for a short duration (35%). In addition, blacks (36.1%), those with income below 15,000$ (33.2%), education below high school (34.7%), and those who had one comorbidity (34.9%) were more likely to use medication for a shorter (≥30 days) duration.

The distribution of the number of prescriptions is shown in Figure 1, in which about 25% of patients received only one prescription. The median number of antidepressant prescriptions was three (IQR: 1.0–42.0). About 43% of prescriptions had days of supply for 30 days, and 50% for 90 days (these numbers are based on the days of supply on all prescriptions). The most commonly prescribed medications were Sertraline (22%), Trazodone (19%), Escitalopram (14%), Citalopram (14%), and Duloxetine (12%).

### 3.2. Resumption of Antidepressant Medications

The patterns of antidepressant resumption among 1084 older adults are presented in Table 2. Overall, 496 (45.8%) patients resumed medication at some point during the follow-up period, and 403 (37.2%) patients resumed within three months, 51 (4.7%) patients between three to six months, and 42 (3.9%) patients resumed after six months. In addition, those who were younger, black, with lower income, less education, married, or had more comorbidities were more likely to resume medications.

The detailed patterns of medication resumption by the continuous durations of antidepressants use are shown in Figure 2. Among those patients who continued their medications for 31–90 days or 90–180 days, the rates of resumption were higher (54.0% and 55.1%, respectively) than those patients who discontinued their medication within 30 days (34.2%). Most of these resumptions occurred within three months (37%).

We employed multivariate Cox proportional hazard models to analyze the time to resumption where all covariables were adjusted in the model (Table 3). We used 91–180 days as a reference duration as it is the recommended duration for the treatment of depression in the guideline [5]. Patients who continued medication for less than 30 days were half as likely to resume the medication compared to those who continued medication for the recommended 91–180 days (hazard ratio (HR): 0.49, 95% CI: 0.37–0.65). In the multivariate logistic model, patients with <30 days of continuous medication were less likely to resume medication within three months (Odds Ratio (OR): 0.36, 95% CI: 0.23–0.54), compared with patients with recommended durations. There were no significant differences in resumption rates by gender, age group, race, education, marital status, or comorbidities.

## 4. Discussion

Although older adults in the US have comparatively lower rates of depression than young or middle age groups [26], about one-third of older adults with depression who received antidepressants had an antidepressant prescription filled for 30 days or less, and it is likely that some of them might have never refilled their medication again, as they were less likely to resume medications compared to those with 91–180 days of continuous duration. Overall, approximately half of the patients refilled medication within three months and one-third after six months or more.

Our findings revealed some conflicting issues. A large proportion of patients with depression only received medications for a short duration, contradicting the APA guidelines. However, almost half of these short-term users did not resume medications in the follow-up, indicating that either short-term antidepressants were effective in relieving the symptoms, or the short-term use was not necessary. Our findings aligned with prior research which found that about 40% of antidepressant users discontinue within the first 30 days of antidepressant prescription [16]. On the other hand, our findings disagreed with previous findings in which short-term users had a higher risk of relapse or recurrence than those who continued their medication regimen for the long term [18,19]. However, these studies were in a different geographic location and among different age groups. Nonetheless, some studies also found no increased risk of relapse or resumption among short-term users after addressing potential biases such as immortal and neglected time bias [12]. Our study was conducted among older US adults, which is different from previous studies in terms of populations and geolocations; thus, it may not be comparable with previous studies.

Older adults may end the antidepressant treatment prematurely for a myriad of reasons, for example, due to the lack of efficacy, experiencing adverse side effects, interaction with other drugs, or temporary relief of depressive symptoms [27]. Notably, not all patients respond positively to antidepressant medications and it may take approximately seven weeks to achieve remission [28]. The APA guidelines recommend 28 days of medication to assess the effectiveness of a prescribed antidepressant [5]. In our study, about 12% of older adults had a medication for less than 30 days but resumed medication within 30 days, presumably resulting from a drug switch or other medical reasons.

On the other hand, many of the short-term users could have experienced mild depressive symptoms, a brief episode of depression, or a comorbid with varied health conditions such as myocardial infarction, stroke, cancer, and diabetes [29,30,31,32,33]. Approximately 80% of our study patients had two or more comorbidities, suggesting that once patients felt a relief of the depressive symptoms, they stopped refilling prescriptions with or without physician’s recommendations.

Serotonergic antidepressants are often used for managing other health conditions. For example, insomnia, pain, and weakness are very common symptoms of depression among older adults [34]. The most common prescribed drugs in our study were Sertraline, Trazodone, Escitalopram Oxalate, and Citalopram Hydrobromide. Trazodone was more likely to be used by short-term users (≤30 days and 31–90 days), whereas other drugs were mainly used by longer-duration patients. Trazodone is often prescribed to treat insomnia in addition to depression [22]. These short-duration treatments were probably for treating some specific symptoms of depression or other related symptoms, and not for major depression, and were prescribed for a shorter period.

Additionally, many older patients are reluctant to accept a diagnosis of a mental disorder, specifically of depression. The diagnosis of mental disorders is associated with a stigma which is significant in older patients as well [35]. In addition, many patients, especially older adults, stopped medication after a short period of time. The delayed onset of action of serotonergic antidepressants is one of the factors of the treatment failure rate. The need for continuing the treatment for a long time adds to the burden of treatment and increases the risk of medication nonadherence.

It might be the most alarming if many short-term users resulted from the over-diagnosis or over-treatment of depression, as the risk of resumption among these short-term users was significantly lower than those with 91–180 days of treatment. Antidepressants are beneficial for those who are following the recommended guideline, i.e., three months during the acute phase and six to nine months during the continuous phase [36]. However, previous studies have documented possible over-prescription or inappropriate prescription of antidepressant medications among older adults [7,9], which is a global concern, including in Australia, the UK, and France [10,37,38]. Some studies have shown higher rates of antidepressant prescriptions by primary care physicians than mental health specialists [39,40]. In our study, about two-thirds of the patients with a physician’s diagnosis of depression had a PHQ8 score below 10, suggesting an over-diagnosis of depression and over-prescription of antidepressants on a considerable scale.

Our study has several strengths. First, we focused on community-dwelling older patients for whom research on depression medications was limited. Second, we included all patients having a depression diagnosis within six months of medication or a PHQ-8 score ≥ 10, and excluded patients who took antidepressants only for other purposes such as anxiety, stress, pain, insomnia, and/or other psychotic conditions. The patients in our cohort were likely to take antidepressants specifically for treating depression. Finally, we also limited the cohort to those who newly initiated medications at least six months after enrollment. Although these patients were not likely to be the newly diagnosed patients, the patterns of medications were less likely to be affected by possible prior antidepressant medications.

Our study has some limitations too. First, we did not have information on the reasons for discontinuation. Second, we used the resumption of medication as a measure of recurrence. Patients who did not resume medication may seek alternative treatment such as cognitive behavior therapy or alternative medicine. We did not have detailed information about psychotherapy on the claims. Third, some of the short-term use of antidepressants is likely for treating conditions other than major depression, for example, some conditions related to depression (e.g., trazodone for insomnia) which we were unable to differentiate. In addition, due to the COVID-19 pandemic, the most recent data (2020–2022) used different modes of survey than previous years, and the release of MCBS data was also delayed. Thus, our data were limited to the pre-COVID period, and the sample size was also moderate. We are requesting new data to explore the impact of COVID-19 on antidepressant medications. Finally, we used claim data for medications to calculate duration, gap, and resumption. Patients who filled their prescriptions might not take their medications for the prescribed time, resulting in the misclassification of treatment duration and resumption.

## 5. Conclusions

Ensuring a sufficient length of antidepressant medications is crucial for the success of controlling moderate to severe depression. It is unsettling that almost one-third of older patients with depression stopped medications in less than 30 days. Although some of them may indeed need treatment for a short duration to manage depressive symptoms, those who resumed medications within a short period were likely to be under-treated, while those without medication resumption indicated an over-prescription of antidepressants. Other treatments such as psychotherapy may be appropriate for patients with mild depression. Therefore, a careful evaluation of patients and a cautious use of medications would reduce the over-prescription of antidepressants and lower the risk of side effects and drug interactions with other medications among older adults.

## Figures and Tables

**Figure 1 ijerph-21-01209-f001:**
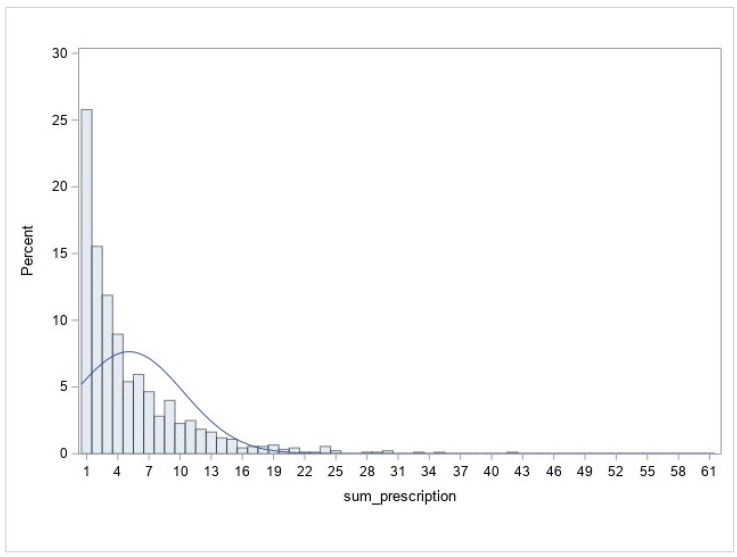
Distribution of the number of prescriptions per patient.

**Figure 2 ijerph-21-01209-f002:**
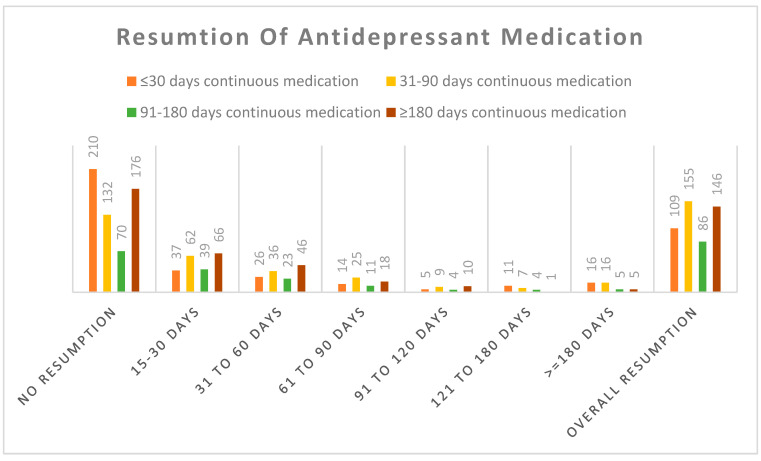
Continuous medication duration and time to resume medication.

**Table 1 ijerph-21-01209-t001:** Demographic information of antidepressant users.

Characteristics			Continuous Treatment Duration		
		≤30 Days	31–90 Days	91–180 Days	≥181 Days
	N (%)	N (%)	N (%)	N (%)	N (%)
Total	1084	319 (29.4)	287 (26.4)	156 (14.3)	322 (29.7)
Gender					
Male	338 (31.2)	108 (31.9)	91 (26.9)	37 (10.9)	102 (30.1)
Female	746 (68.8)	211 (28.2)	196 (26.2)	119 (15.9)	220 (29.4)
Age					
65–74	444 (40.9)	100 (22.5)	118 (26.5)	60 (14.6)	103 (25.1)
75–85	411 (37.9)	139 (33.8)	109 (26.5)	60 (14.6)	103 (25.1)
≥85	229 (21.1)	80 (34.9)	60 (26.2)	30 (13.1)	59 (25.7)
Race					
White	930 (85.7)	269 (28.9)	241 (25.9)	134 (14.4)	286 (30.7)
Black	72 (6.6)	26 (36.1)	22 (30.5)	10 (13.8)	14 (19.4)
Other	82 (7.5)	24 (29.2)	24 (29.2)	12 (14.6)	22 (26.8)
Income					
<15,000	253 (23.3)	84 (33.2)	64 (25.3)	38 (15.0)	67 (26.4)
15,000–<30,000	327 (30.1)	100 (30.5)	88 (26.9)	42 (12.8)	97 (29.6)
30,000–<50,000	202 (18.6)	57 (28.2)	66 (32.6)	30 (14.8)	49 (24.2)
≥50,000	302 (27.8)	78 (25.8)	69 (22.8)	46 (15.2)	109 (36.1)
Education					
Below high school	236 (21.7)	82 (34.7)	59 (25.0)	30 (12.7)	65 (27.5)
High school graduate	288 (26.5)	85 (29.5)	81 (28.1)	32 (11.1)	90 (31.2)
College and above	560 (51.6)	152 (27.1)	147 (26.2)	94 (16.7)	167 (29.8)
Marital status					
Married	623 (57.4)	191 (30.6)	165 (26.4)	86 (13.8)	181 (29.1)
Widowed, divorced, single and other	461 (42.5)	128 (27.7)	122 (26.4)	70 (15.1)	141 (30.5)
Comorbidity					
0–1 comorbidity	226 (20.8)	79 (34.9)	68 (30.1)	29 (12.8)	50 (22.1)
Two comorbidities	221 (20.3)	69 (31.2)	59 (26.7)	31 (14.0)	62 (28.1)
Three comorbidities	212 (19.5)	64 (30.1)	53 (25.0)	39 (18.4)	56 (26.4)
Four or more comorbidities	425 (39.2)	107 (25.1)	107 (25.1)	57 (13.4)	154 (36.2)
Heart disease					
No	577 (53.2)	172 (29.8)	155 (26.8)	82 (14.2)	168 (29.1)
Yes	507 (46.7)	147 (28.9)	132 (26.0)	74 (14.6)	154 (30.3)
Stroke					
No	913 (84.2)	257 (28.1)	253 (27.7)	134 (14.6)	269 (29.4)
Yes	171 (15.7)	62 (36.2)	34 (19.8)	22 (12.8)	53 (30.9)
Diabetes					
No	698 (64.3)	206 (29.5)	187 (26.7)	99 (14.1)	206 (29.5)
Yes	386 (35.6)	113 (29.2)	100 (25.9)	57 (14.7)	116 (30.1)
Cancer					
No	840 (77.4)	252 (30.0)	214 (25.4)	123 (14.6)	251 (29.8)
Yes	244 (22.5)	67 (27.4)	73 (29.9)	33 (13.5)	71 (29.1)

**Table 2 ijerph-21-01209-t002:** Medication resumption of antidepressant users with depression.

Characteristics	Overall Resumption within the Follow-Up	Resumption within Three Months	Resumption between Three to Six Months	Resumption after Six Months
	N (%)	N (%)	N (%)	N (%)
Total (1084)	496 (45.8)	403 (37.2)	51 (4.7)	42 (3.9)
Gender				
Male	144 (42.6)	125 (36.9)	12 (3.6)	7 (2.1)
Female	352 (47.2)	278 (37.3)	39 (5.2)	35 (4.7)
Age				
65–74	216 (48.7)	181 (40.8)	22 (4.95)	13 (2.9)
75–85	185 (45.0)	149 (36.3)	19 (4.6)	17 (4.1)
≥85	95 (41.5)	73 (31.9)	10 (4.4)	12 (5.2)
Race				
White	425 (45.7)	345 (37.1)	45 (4.8)	35 (3.8)
Black	38 (52.8)	33 (45.8)	3 (4.2)	2 (2.7)
Other	33 (40.2)	25 (30.5)	3 (3.7)	5 (6.1)
Income				
<15,000	127 (50.2)	99 (39.1)	12 (4.7)	16 (6.3)
15,000–<30,000	147 (44.9)	112 (34.3)	20 (6.1)	15 (4.6)
30,000–<50,000	100 (49.5)	85 (42.1)	11 (5.4)	4 (1.9)
≥50,000	122 (40.4)	107 (35.4)	8 (2.7)	7 (2.3)
Education				
Below high school	116 (49.2)	90 (38.1)	13 (5.5)	13 (5.5)
High school graduate	136 (47.2)	105 (36.5)	18 (6.3)	13 (4.5)
College and above	244 (43.6)	208 (37.1)	20 (3.6)	16 (2.9)
Marital status				
Married	291 (46.7)	228 (36.6)	30 (4.8)	33 (5.3)
Widowed, divorced, single and other	205 (44.5)	175 (37.9)	21 (4.6)	9 (1.9)
Comorbidity				
No-1 comorbidity	88 (38.9)	77 (34.1)	6 (2.6)	5 (2.2)
Two comorbidities	103 (46.6)	87 (39.4)	5 (2.3)	11 (4.9)
Three comorbidities	95 (44.8)	77 (36.3)	12 (5.7)	6 (2.8)
Four or more comorbidities	210 (49.4)	162 (38.1)	28 (5.6)	20 (4.7)
Heart disease				
No	259 (44.9)	228 (39.5)	14 (2.4)	17 (2.95)
Yes	237 (46.8)	175 (34.5)	37 (7.3)	25 (4.9)
Stroke				
No	413 (45.2)	342 (37.5)	39 (4.3)	33 (3.5)
Yes	83 (48.5)	61 (35.7)	12 (7.0)	10 (5.9)
Diabetes				
No	323 (46.3)	258 (36.9)	36 (5.2)	29 (4.2)
Yes	173 (44.8)	145 (37.6)	15 (3.8)	13 (3.4)
Cancer				
No	381 (45.4)	311 (37.0)	39 (4.6)	31 (3.6)
Yes	115 (47.1)	92 (37.7)	12 (4.9)	11 (4.5)

**Table 3 ijerph-21-01209-t003:** Cox proportional hazard ratio and Odds ratio of resumption of medication.

Characteristics	Overall Resumption	Resumption within Three Months	Resumption within Six Months	Resumption after Six Months
	HR (95% CI)	OR (95% CI)	OR (95% CI)	OR (95% CI)
Continuous treatment				
91 to 180 days	ref.	ref.	ref.	
≤30 days	0.49 (0.37, 0.65)	0.36 (0.23, 0.54)	1.01 (0.41, 2.45)	0.63 (0.22, 1.80)
31 to 90 days	0.91 (0.69, 1.19)	0.84 (0.56, 1.26)	1.07 (0.44, 2.61)	0.53 (0.18, 1.52)
>181 days	0.76 (0.58, 0.99)	0.77 (0.52, 1.14)	0.58 (0.22, 1.52)	2.26 (0.63, 8.07)
Gender				
Male	ref.	ref.	ref.	
Female	1.09 (0.89, 1.30)	0.99 (0.74, 1.31)	1.52 (0.76, 3.03)	0.56 (0.24, 1.31)
Age				
65–74	ref.	ref.	ref.	
75–85	0.90 (0.74, 1.10)	0.89 (0.67, 1.19)	0.77 (0.40, 1.50)	0.92 (0.42, 1.99)
≥85	0.79 (0.62, 1.03)	0.49 (0.51, 1.06)	0.75 (0.33, 1.68)	0.86 (0.36, 2.03)
Race				
White	ref.	ref.	ref.	
Black	1.07 (0.76, 1.51)	1.39 (0.83, 2.32)	0.68 (0.19, 2.35)	2.23 (0.50, 9.92)
Other	0.69 (0.48, 1.01)	0.63 (0.37, 1.09)	0.62 (0.17, 2.22)	0.97 (0.33, 2.81)
Income				
<15,000	ref.	ref.	ref.	
15,000–<30,000	0.84 (0.65, 1.09)	0.77 (0.53, 1.12)	1.28 (0.57, 2.88)	1.24 (0.55, 2.79)
30,000–<50,000	1.03 (0.76, 1.39)	1.04 (0.67, 1.62)	1.14 (0.43, 3.01)	2.48 (0.73, 8.04)
≥50,000	0.78 (0.57, 1.07)	0.75 (0.48, 1.18)	0.64 (0.21, 1.94)	1.46 (0.47, 4.51)
Education				
Below high school	ref.	ref.	ref.	
High school graduate	0.90 (0.69, 1.17)	0.87 (0.59, 1.28)	1.13 (0.52, 2.48)	0.92 (0.39, 2.16)
College and above	0.81 (0.63, 1.04)	0.87 (0.60, 1.26)	0.68 (0.30, 1.53)	1.31 (0.55, 3.11)
Marital status				
Married	ref.	ref.	ref.	
Widowed, divorced, single and other	0.97 (0.78, 1.19)	0.94 (0.70, 1.26)	0.79 (0.41, 1.53)	0.51 (0.21, 1.21)
Comorbidity				
No-1 comorbidity	ref.	ref.	ref.	
Two comorbidities	1.29 (0.97, 1.72)	1.23 (0.83, 1.83)	0.91 (0.27, 3.05)	0.42 (0.14, 1.25)
Three comorbidities	1.22 (0.91, 1.64)	1.05 (0.71, 1.58)	2.22 (0.81, 6.09)	0.77 (0.22, 2.63)
Four or more comorbidities	1.31 (1.01, 1.68)	1.10 (0.77, 1.56)	2.65 (1.06, 6.61)	0.46 (0.16, 1.27)

## Data Availability

Data are from the Medicare Current Beneficiary Survey, which is available here https://www.cms.gov/data-research/research/medicare-current-beneficiary-survey accessed on 12 September 2024.

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
