# Peer review of "Discontinuation of Antidepressants and the Risk of Medication Resumption among Community-Dwelling Older Adults with Depression in the US"

_ijerph, 2024, doi:10.3390/ijerph21091209_

Round 1

Reviewer 1 Report

Comments and Suggestions for Authors

Thank you for providing the opportunity to review this manuscript. The paper reads well, but requires some revisions as stated below.

Introduction:

Line 81 and 93, fall should be with capital 'F', Fall, and there is a typing error line 93.

Methods:

Line 105 - 107 should be included in the 'Introduction' section of the paper, as it discusses previous literature.

What was the rationale for using the cut off of 14 days for defining continuous duration and resumption of medication. Was this based on a priori literature or some other existing guidelines, if so, please add a reference.

Recommendation: It would be interesting to add a supplementary table on sensitivity analysis performed on a subgroup of participants with neuropsychic disease and how the pattern of prescription differed in those group of patients. 

Results:  

Table 1 & 2: There seems to be some data missed with the categorization of age and income variable. Please list the N (%) for categories of age as <=65 to <=75; >75 to <=85, and >85 years. Similarly  the categorization for income variable seems to have an overlap with the income level of 30,000 counted twice as 15,000 - 30,000 and 30,000 - 50,000 please ensure, no data is lost and there is no overlap in between the categories. 

If available, please add information on ethnicity.

Figure 1 is not warranted. Instead recommend to include a bar graph showing the different types of prescription medications, and a box plot for IQR, especially because the authors have discussed the different types of drugs in the discussion section as well. 

Discussion: Line 238 - 242

Did the studies with which your findings differed use the same age group of people, for reference 19 the study was conducted in Korea, so the results may be different based on geographic location or some other factors. Please discuss, in what aspects did your study results differ from the recent research on the same topic.

Line 228, 246, 255, 266: Recommend to avoid using words like 'might' and 'may' in the discussion section. The discussion should be written in an affirmative tone, stating the facts and elaborating the reasons for your study findings with the support of literature, furthering it citing literature where similar results were found in the past. At present it reads like the authors are not sure about the remissions, reasons for not refilling the medications or why participants stopped refilling prescriptions. 

Line 276: Instead of 'this MIGHT suggest' recommend to paraphrase it as ' A PHQ8 score of <10 suggests an over diagnosis of..' Please refrain from using the word 'might' in every sentence as it shows the authors are not confident about the study results and are merely guessing it, although that may not be true.

Line 285: Possibly rephrase as 'Although these patients were not likely to be newly diagnosed, the patterns were...'. Delete the repetitive use of the word 'might'. 

The manuscripts needs to be reviewed by authors for style and impact of writing when presenting and discussing results. This is an important topic which could be of interest to some researchers. 

Comments on the Quality of English Language

The manuscripts needs to be reviewed by authors for style of writing and avoid using repetitive words. Writing needs to be in a more assertive tone. 

Author Response

Thank you for providing the opportunity to review this manuscript. The paper reads well, but requires some revisions as stated below.

Introduction:

Line 81 and 93, fall should be with capital 'F', Fall, and there is a typing error line 93.

Response: Thank you for your comment. We have corrected line 81 and 93.

Methods:

Line 105 - 107 should be included in the 'Introduction' section of the paper, as it discusses previous literature.

Response: Many thanks for this. They  were moved to the introduction section in line 74-76.

What was the rationale for using the cut off of 14 days for defining continuous duration and resumption of medication. Was this based on a priori literature or some other existing guidelines, if so, please add a reference.

Response: According to American Psychiatric Association, recommended antidepressants use duration is at least 4-6 weeks. If the drug is not effective, then the physician can switch to another drug from same class or different class. It is possible that patient may switch to different drugs. The washout period is 1-2 weeks for most serotonergic drugs. That’s the reason we used an interval of 14 days cut off for continuous duration and relapse. I have added a reference in line -120.

Recommendation: It would be interesting to add a supplementary table on sensitivity analysis performed on a subgroup of participants with neuropsychic disease and how the pattern of prescription differed in those group of patients. 

Response: We did not include patients with neuropsychic diseases in the analytic data. In fact, we specifically included only those with a depression diagnosis or depressive symptoms so that the analysis is more specific to those with depression.  However, this suggestion is an interesting issue we will address in the future.

Results:  

Table 1 & 2: There seems to be some data missed with the categorization of age and income variable. Please list the N (%) for categories of age as <=65 to <=75; >75 to <=85, and >85 years. Similarly  the categorization for income variable seems to have an overlap with the income level of 30,000 counted twice as 15,000 - 30,000 and 30,000 - 50,000 please ensure, no data is lost and there is no overlap in between the categories. 

Response:  Many thanks for this. There is a little bit confusion in the writing. The age was categorized as 65-74, 75-84, ≥85. The income was categorized as 15,000 - <30,000; 30,000 - <50,000; ≥50,000. The problem was in the writing which I corrected. There is no missing data. We have fixed the table accordingly.

If available, please add information on ethnicity.

Response: Thank you for suggestion. It would be great to add ethnicity but we do not have the detail information on ethnicity for all patients.

Figure 1 is not warranted. Instead recommend to include a bar graph showing the different types of prescription medications, and a box plot for IQR, especially because the authors have discussed the different types of drugs in the discussion section as well. 

Response: Thank you for the suggestion. We have another research paper specifically on different medications. Therefore, we will keep the figure as is, and leave the detailed analysis in another paper.  

Discussion: Line 238 - 242

Did the studies with which your findings differed use the same age group of people, for reference 19 the study was conducted in Korea, so the results may be different based on geographic location or some other factors. Please discuss, in what aspects did your study results differ from the recent research on the same topic.

Response: Our population and sample are different from those of other studies (e.g., we are exclusively analyzing elderly people). So, we got a different result. We have made some corrections in line 235-241.

Line 228, 246, 255, 266: Recommend to avoid using words like 'might' and 'may' in the discussion section. The discussion should be written in an affirmative tone, stating the facts and elaborating the reasons for your study findings with the support of literature, furthering it citing literature where similar results were found in the past. At present it reads like the authors are not sure about the remissions, reasons for not refilling the medications or why participants stopped refilling prescriptions.

Response: Thank you again. This is a great suggestion. we have rewritten the sentences without might/may. However, we would avoid being too assertive as our study won’t prove causative associations. 

Line 276: Instead of 'this MIGHT suggest' recommend to paraphrase it as ' A PHQ8 score of <10 suggests an over diagnosis of..' Please refrain from using the word 'might' in every sentence as it shows the authors are not confident about the study results and are merely guessing it, although that may not be true.

Response: Thanks. This sentence is corrected in line 275.

Line 285: Possibly rephrase as 'Although these patients were not likely to be newly diagnosed, the patterns were...'. Delete the repetitive use of the word 'might'. 

Response: This sentence is corrected in line 285.

The manuscripts needs to be reviewed by authors for style and impact of writing when presenting and discussing results. This is an important topic which could be of interest to some researchers. 

Comments on the Quality of English Language

The manuscripts needs to be reviewed by authors for style of writing and avoid using repetitive words. Writing needs to be in a more assertive tone. 

Response: Many thanks for your suggestions. We have re-read the whole text by another author to correct small language issues.

Reviewer 2 Report

Comments and Suggestions for Authors

Hello, dear colleagues!

Thank you for your interesting research, but I have a few follow-up questions.

1. Why was the survey conducted between 2015 and 2019, and you are preparing the article only in 2024? What restrictions were there in publishing fresh data?

2. Do you think it would be appropriate to talk about a significant difference in Predictors between people who have not been ill and those who have had Covid-19?

3. How was the sample size of participants calculated?

4. What is your null hypothesis?

5. The results are too specific and look like a division into races, where whites and blacks are segregated from others. Try to give a more comprehensive presentation first, and then condense it later.

6. The list of references contains references that have lost their relevance over a period of time. Perhaps they were useful for the period of the study, but being 10 or more years old does not allow them to be relevant. It is necessary to replace the sources and take this into account in the text

Author Response

Thank you for your interesting research, but I have a few follow-up questions.

Response: Many thanks for your time and help.

  1. Why was the survey conducted between 2015 and 2019, and you are preparing the article only in 2024? What restrictions were there in publishing fresh data?

Response: Thank you for your question. This is a secondary analysis. We used data from Medicare Current Beneficiary Survey data (MCBS). The data were updated to 2019, and due to COVID-19 during the years 2020-2022, the survey instruments were modified and data were different from previous surveys, and it is well known that health care use during the COVID-19 period is different as well.

  1. Do you think it would be appropriate to talk about a significant difference in Predictors between people who have not been ill and those who have had Covid-19?

Response: This is a great question and we are requesting new data for the year 2020-2022 to explore these issues (data were released late due to COVID-19 too).

  1. How was the sample size of participants calculated?

Response: As a secondary data analysis, we are limited by the available data. On the other hand, we had about similar prevalence of depression (10%) as that of general US population, and our analytic sample size is above 1000, which is sufficient for descriptive analysis in this study. It is not a big data analysis, but sufficiently large for simple statistical testing.   

  1. What is your null hypothesis?

Response: as a descriptive analysis, we state the alternative hypothesis in the text. The implicit null hypothesis is that the duration of antidepressant users is not related to the resumption of medication.

  1. The results are too specific and look like a division into races, where whites and blacks are segregated from others. Try to give a more comprehensive presentation first, and then condense it later.

Response: This is a good observation. We have presented the overall rates in the first row of tables 1,2,3. On the other hand, As our data are based on the US survey, it is always presented by race (Blacks, Whites, and others), as racial indicator is often an indicator of socio-economic status.

  1. The list of references contains references that have lost their relevance over a period of time. Perhaps they were useful for the period of the study, but being 10 or more years old does not allow them to be relevant. It is necessary to replace the sources and take this into account in the text

Response: Many thanks for this insightful observation. We have noticed the gap in the literature regarding mental health and medications among elderly people and recognized that this field was not well studied. Therefore, we included several older references, and believe that it is time for an update for this elderly population.

Reviewer 3 Report

Comments and Suggestions for Authors

This is an interesting paper, identifying the gap between ideal and actual usage of a group of drugs. Practice guidelines and text books apart, actual medical practices often deviate from recommended practices. This happened with many drugs, and when it happens with the most commonly prescribed agents, viz antidepressants, it is a matter of concern.

It is well known that serotoninergic antidepressants usually have an onset of action of between 3 to 6 weeks and show their full effect by around 12 weeks. When used for shorter periods they may not provide the intended benefits.

Nonetheless in actual practice drugs are used at doses and schedules far different from those recommended. This paper explores the use of antidepressants in a vulnerable age group and attempts to correlate the duration of usage with different factors such as age, income, marital status and co morbidities. The authors have observed and documented the gap, which could be because of a variety of reasons, starting from inability to understand the doctors instructions to actual usage by patients above the age of 65.

1. The incidence of depression is quite high, even given the uncertainties in the diagnosis of depression. Despite the availability of DSM, very few practising doctors actually follow the DSM before diagnosing it, at least in my country. Elsewhere too, there are fears that depression is either underdiagnosed or overdiagnosed. 

2. There is reluctance on the part of some patients to accept a diagnosis of depression. This reluctance exists for a variety of mental disorders, and though it is waning, the diagnosis of mental disorders is associated with a stigma. I suspect this is significant in older patients.

3. Among the variety of drugs available today for the treatment of mental disorders, antidepressants have a high failure rate. One of the reasons is the unduly delayed onset of action of some of the commonly used ones. The need for continuing the treatment for a fairly long time adds to the failure rate.

4. Many patients give up the drug shortly after starting it, (the current paper shows that it could be as high as 30%) only to start it again, while some just drop out. In old age homes, absence of family makes it difficult to ensure that they continue using the drug as prescribed. Of course, my knowledge about old age home conditions is rather limited, for in my country, putting a parent in such a facility is considered a sign of poor upbringing. Well brought up people, never never put their parents in such facilities (this is in our country, I am not passing any opinion on what happens elsewhere).

5.If I were to do this research I would also study the factors that I think would affect drug taking behaviour, incidentally I would have studied the same factors as the current authors.

6. It is for these reasons that I felt that the antidepressant drug treatment needs to be studied. I feel factors like gender, race, age, financial status, marital status are likely to have a significant impact on the duration for which the medication is continued and need to be studied, and that is what the authors have done.

7. I would have appreciated it if the authors had analysed deeper into how the presence or absence of a spouse, financial conditions or comorbidities  affects the compliance to medication, but given the fact that the data was not obtained from personal interviews, makes such analyses difficult. Given the data source, I think the authors have done the best that they could.

Author Response

This is an interesting paper, identifying the gap between ideal and actual usage of a group of drugs. Practice guidelines and text books apart, actual medical practices often deviate from recommended practices. This happened with many drugs, and when it happens with the most commonly prescribed agents, viz antidepressants, it is a matter of concern.

It is well known that serotoninergic antidepressants usually have an onset of action of between 3 to 6 weeks and show their full effect by around 12 weeks. When used for shorter periods they may not provide the intended benefits.

Response: total agree.

Nonetheless in actual practice drugs are used at doses and schedules far different from those recommended. This paper explores the use of antidepressants in a vulnerable age group and attempts to correlate the duration of usage with different factors such as age, income, marital status and co morbidities. The authors have observed and documented the gap, which could be because of a variety of reasons, starting from inability to understand the doctors instructions to actual usage by patients above the age of 65.

Response: Agree and thanks for your comments.

  1. The incidence of depression is quite high, even given the uncertainties in the diagnosis of depression. Despite the availability of DSM, very few practising doctors actually follow the DSM before diagnosing it, at least in my country. Elsewhere too, there are fears that depression is either underdiagnosed or overdiagnosed. 

Response: Thank you for your comment. We also recognized that mental health among elderly population in the US somehow is neglected. That is the reason this study is important and we have mentioned this issue in our discussion.

  1. There is reluctance on the part of some patients to accept a diagnosis of depression. This reluctance exists for a variety of mental disorders, and though it is waning, the diagnosis of mental disorders is associated with a stigma. I suspect this is significant in older patients.

Response: Thank you for this important issue. We have discussed this in discussion line 265-271.

  1. Among the variety of drugs available today for the treatment of mental disorders, antidepressants have a high failure rate. One of the reasons is the unduly delayed onset of action of some of the commonly used ones. The need for continuing the treatment for a fairly long time adds to the failure rate.

Response: Agree and thanks for your suggestions. We have discussed this in discussion line 265-271.

  1. Many patients give up the drug shortly after starting it, (the current paper shows that it could be as high as 30%) only to start it again, while some just drop out. In old age homes, absence of family makes it difficult to ensure that they continue using the drug as prescribed. Of course, my knowledge about old age home conditions is rather limited, for in my country, putting a parent in such a facility is considered a sign of poor upbringing. Well brought up people, never never put their parents in such facilities (this is in our country, I am not passing any opinion on what happens elsewhere).

Response: Many thanks for your comments from a different perspective. Even within the US, acknowledging the disease and seeking care is not satisfactory. There were too many pushes and pulls in the community and also in the healthcare systems. We have discussed this in discussion line 265-271.

5.If I were to do this research I would also study the factors that I think would affect drug taking behaviour, incidentally I would have studied the same factors as the current authors.

Response: Thank you for your support. We have several topics laid out down the road, but this healthcare use patterns and healthcare seeking behavior are a first step to understanding these important issues.

  1. It is for these reasons that I felt that the antidepressant drug treatment needs to be studied. I feel factors like gender, race, age, financial status, marital status are likely to have a significant impact on the duration for which the medication is continued and need to be studied, and that is what the authors have done.

Response: totally agree. There are many factors that affect the pattern of antidepressant use among elderly people. Some of the socio-economic factors may not be available in our data, but we agree that these are the first and important issues to study. We have added these comments in the discussion section (as limitations and future research).  

  1. I would have appreciated it if the authors had analysed deeper into how the presence or absence of a spouse, financial conditions or comorbidities  affects the compliance to medication, but given the fact that the data was not obtained from personal interviews, makes such analyses difficult. Given the data source, I think the authors have done the best that they could.

Response: Thank you for your comment. We have used marital status as a covariable which indicates if they were living with spouse or without a spouse/partner. We also have income of the beneficiaries in the study. Same as comorbidities. We did not find any significant differences among those groups. However, we would like to explore further in these areas and also agree with the reviewer that socio-economic status in addition to other comorbidities have important implications in antidepressant use, especially among elderly people. These will be our future research topics.

Round 2

Reviewer 2 Report

Comments and Suggestions for Authors

Dear authors!

Despite the fact that you used on the US survey, I believe that such obvious racial segregation in scientific literature is unacceptable

I leave it to the editor's decision, but I believe this point is appropriate for correction

Response: Racial disparity exists in health and healthcare in the US, and our results suggest that such disparities in anti-depressant use were possible, and we did not intend to ignore them.